# Management of Intrahepatic Cholangiocarcinoma

**DOI:** 10.3390/jcm10112368

**Published:** 2021-05-27

**Authors:** Sudha Kodali, Akshay Shetty, Soumya Shekhar, David W. Victor, Rafik M. Ghobrial

**Affiliations:** 1Sherrie and Alan Conover Center for Liver Disease and Transplantation, Houston, TX 77030, USA; skodali@houstonmethodist.org (S.K.); Ashetty@houstonmethodist.org (A.S.); RMGhobrial@houstonmethodist.org (R.M.G.); 2Houston Methodist Research Institute, Houston, TX 77030, USA; 3Texas A&M College of Medicine, Houston Campus, Houston, TX 77030, USA; sshekhar@exchange.tamu.edu

**Keywords:** cholangiocarcinoma, intrahepatic, resection, chemotherapy, transarterial chemoembolization (TACE), Yttrium-90 (Y90), chemotherapy, short-beam external radiation, proton beam therapy, brachytherapy, transplantation

## Abstract

Cholangiocarcinoma is a tumor that arises as a result of differentiation of the cholangiocytes and can develop from anywhere in the biliary tree. Subtypes of cholangiocarcinoma are differentiated based on their location in the biliary tree. If diagnosed early these can be resected, but most cases of intrahepatic cholangiocarcinoma present late in the disease course where surgical resection is not an option. In these patients who are poor candidates for resection, a combination of chemotherapy, locoregional therapies like ablation, transarterial chemo and radioembolization, and in very advanced and metastatic disease, external radiation are the available options. These modalities can improve overall disease-free and progression-free survival chances. In this review, we will discuss the risk factors and clinical presentation of intrahepatic cholangiocarcinoma, diagnosis, available therapeutic options, and future directions for management options.

## 1. Introduction

Cholangiocarcinoma (CCA) is the second most common primary liver cancer and has been increasing in incidence globally [1]. This tumor arises from the biliary epithelial cells and patients usually present with advanced disease and hence it is associated with a high mortality rate [2]. Cases of CCA are classified based on area of origin in the biliary tree and can be divided into distal CCA, hilar, and intrahepatic CCA and cases of the intrahepatic CCA can be further classified into three subtypes: (i) mass-forming tumors, (ii) periductal-infiltrating type, and (iii) intraductal based on their growth pattern [3,4]. Figure 1 illustrates the three subtypes of intrahepatic cholangiocarcinoma (iCCA).

The incidence of iCCA has been increasing, as shown in many studies, but was confirmed in a large surveillance, epidemiology, and end results (SEER) database review showing a 128% increase in the last 40 years with significant increase in the last 10 years, including increase in relative incidence in men and Hispanics [1,5]. These increased rates are felt to be secondary to early diagnosis because of improved diagnostic modalities in addition to changes in trends of epidemiological and environmental factors [1]. CCA sometimes develops sporadically. but otherwise choledochal cysts, primary sclerosing cholangitis, biliary cirrhosis, cholelithiasis, thorotrast exposure, Caroli’s disease, hepatobiliary flukes, and cirrhosis of the liver irrespective of the etiology are the risk factors associated with development of intrahepatic CCA and are summarized in Table 1 [6,7,8,9]. In the United states and other developed countries, non-alcoholic steatohepatitis (NASH) may account for increasing incidence of cholangiocarcinoma [10]. Patients with NASH-related iCCA have been shown to have low overall survival rates compared to patients without known liver disease but have similar survival rates to that of patients with other risk factors for cirrhosis [11]. All these risk factors cause chronic inflammation secondary to damage of biliary epithelium and bile stasis, and there are several mechanisms that have been proposed to be the pathways for cholangiocarcinogenesis [12]. Some of them include production of growth factors, cytokines, upregulation of oncogenesis, mutations in cell cycle genes, DNA damage, high levels of telomerase messenger RNA (mRNA), and vascular endothelial growth factors [13,14]. These patients in general do poorly with a median survival of 12–24 months without treatment [15]. If patients present early, surgery can offer a 5-year survival rate of 15–40% [16]. In patients who are non-resectable, chemotherapy along with locoregional modalities and radiation help improve survival. Molecular profiling has improved our understanding of actionable mutations in the last decade and the genes that have been reported in iCCA are isocitrate dehydrogenase (IDH)1, IDH2, fibroblast growth factor receptor (FGFR)1, FGFR2, FGFR3, epoxide hydrolase (EPH)A2, and biofilm-associated surface protein (BAP)1 genes [17].

## 2. Diagnosis

Intrahepatic CCA can be diagnosed during routine hepatocellular carcinoma screening and surveillance in patients with cirrhosis, and these patients can be completely asymptomatic when present with small tumors. However, a larger proportion of patients may not have cirrhosis or pre-existing liver disease and may present with advanced disease and symptoms like jaundice, abdominal pain, anorexia, and weight loss and imaging often reveals a mass in the liver [18]. Contrast-enhanced triple-phased cross-sectional imaging, which includes arterial, portal, and hepatic venous phases, in combination with biopsy is often needed for a definitive diagnosis [19]. The contrast-enhanced imaging not only helps identify the location, size of the mass, and presence or absence of satellite lesions, but can also provide adequate information regarding vascular invasion, lymphadenopathy, and liver volumetric assessments to determine the eligibility for surgical resection [20]. Unlike hepatocellular carcinoma (HCC), where the LI-RADS system offers definitive diagnosis of HCC based on the imaging characteristics, such as arterial phase enhancement, venous washout, and presence of a pseudocapsule, iCCA is difficult to diagnose and differentiates from HCC, especially tumors with a mixed hepatocellular-cholangiocarcinoma component [19]. Intrahepatic CCA can be diagnosed based on the pattern of enhancement on gadoxetic-acid-enhanced MRI, where a peripheral rim with internal heterogenous enhancement and the hepatobiliary phase with smaller satellite nodules aid in diagnosis [21]. A positron emission tomography (PET) scan is used for staging and to rule out extrahepatic metastasis, and as iCCA is a fluoro-2-deoxy-D-glucose (FDG) avid tumor in the majority of cases, a combination of PET and computed tomography (CT) scans increases the sensitivity and diagnostic yield to about 90% [22,23]. Image- or endoscopy-guided biopsy, though occasionally challenging because of the intense desmoplasmic reaction around the neoplasm, helps confirm the diagnosis and provides valuable information regarding actionable mutations via molecular profiling [24]. Fluorescence in situ hybridization (FISH) analysis is used in addition to microscopic examination of the obtained tissue to aid in the diagnosis [25]. Carbohydrate antigen 19-9 (CA19-9) measurement has limited utility if the levels are mildly elevated as they are often seen in PSC and other cholestatic conditions, but a CA19-9 value greater than 1000 U/mL has higher sensitivity and specificity and is suggestive of larger tumor burden [6,26,27].

Our understanding of tumor genetics has slowly improved in the last couple of decades, but the genetics of CCA remain incompletely understood potentially due to its lower incidence. The role of circulating tumor DNA (ctDNA), which is tumor genetic material that is secreted into the blood, has been extensively explored in the last five years in CCA, and with the current commercially available tissue-based assays, targetable mutations are easier to identify to help guide therapy [28,29,30,31]. The advent of ctDNA may help with early detection of iCCA in asymptomatic patients, even in those with smaller tumors that may not be seen on imaging studies [32].

## 3. Treatment

Treatment options for iCCA include surgical resection, locoregional therapies, chemotherapy and radiation, and liver transplantation. It is very important to note that resection is the most commonly used approach for a cure, though liver transplant is also being considered in some centers on a selective basis based on recent data with improved outcomes. Figure 2 summarizes our recommended approach to management of iCCA, especially the protocol followed at our center when patients receive a new diagnosis of iCCA.

### 3.1. Surgical Resection

Complete surgical resection with negative margins (R0) for smaller tumors is feasible if diagnosed early and offers excellent post-surgical outcomes; 3-year overall survival (OS) was 52–68.6% in R0 resections, but surgery is unfortunately not an option for patients with cirrhosis and portal hypertension or for larger tumors with vascular invasion [33,34,35]. Liver transplantation (LT) may be an option in patients with cirrhosis, but it is offered in limited centers in the USA. Factors that are associated with poor prognosis in iCCA include invasion of liver capsule, positive surgical margins, regional LN metastases, mass-forming or periductal-infiltrating type CCA, and elevated perioperative CA19-9 level greater than 1000 U/mL [15]. Pawlik et al. looked at the impact of surgical margin status on long-term outcomes after surgical resection and showed that in patients with R0 resection there was an incremental worsening of recurrence-free survival (RFS) and overall survival as the resection margin width decreased from ≥1 cm versus 5–9 mm versus 1–4 mm, and hence it is recommended to obtain at least a 1 cm margin width to improve long-term outcomes [36]. Patients who have a recurrence after resection can also be re-explored for repeat resection as shown in a study from Germany where patients who had a repeat resection after recurrence had a median OS of 65.2 months and a 5-year OS rate of 62%, highlighting the importance of re-exploration as an option if technically possible [37].

### 3.2. Liver Transplant

Complete resection of iCCA remains the treatment of choice with improved survival and lower rates of disease recurrence [38,39]. However, due to delayed presentation with advanced disease, patients are often poor candidates for surgical resection and in these patients a complete hepatectomy followed by orthotopic liver transplantation (OLT) is pursued with a curative intent. However, unlike surgical resection, liver transplantation for intrahepatic cholangiocarcinoma remains controversial. Studies from the last two decades of the 20th century revealed poor survival rates of ~30% at 2 years and high recurrence rates (70–84%), but these studies lacked granularity in distinguishing between different BTCs and tumor burden [40,41]. A single-center retrospective analysis from the University of California Los Angeles (UCLA) of 25 patients with iCCA, spanning from 1985 to 2009, compared OLT against resection and showed better recurrence-free survival (RFS) at 3 and 5 years for patients who received a transplant [42]. A retrospective analysis from a single center showed that out of 681 patients transplanted with a presumed diagnosis of primary liver cancer 44 were found to have iCCA on explant, of which 12 met the criteria for smaller tumors or early CCA, and their 5-year overall survival was 63.6 versus 70.3% [43]. Another Spanish multicenter retrospective analysis of 27 patients, who were transplanted with a presumed diagnosis of hepatocellular cancer (HCC) but on explant had iCCA, revealed good overall survival with a 5-year survival of 73% seen in patients with single tumors ≤ 2 cm in size, supporting further the idea of transplanting a carefully selected group of patients with small iCCA [44]. An international consortium of 17 centers that included patients transplanted for decompensated liver disease or HCC but on explant were found to have incidental iCCA or misdiagnosed iCCA included 15 (31%) patients with very early iCCA (single tumors ≤ 2 cm) versus 33 (69%) patients with advanced iCCA (single tumors > 2 cm or multifocal) and found that 1-, 3-, and 5-year survival rates were 93%, 84%, and 65% in the very early iCCA group compared to 79%, 50%, and 45% in the advanced iCCA group, respectively, again showing that transplant for small cholangiocarcinomas is associated with good outcomes [45].

These studies contribute to growing evidence in favor of liver transplantation to treat iCCA and highlights the importance of developing a patient selection criterion. Lunsford et al., from our institution, reported a prospective case series of 6 patients who underwent OLT, out of 21 who were evaluated, after neoadjuvant chemotherapy for biopsy-proven locally advanced iCCA without extrahepatic disease or vascular invasion, and showed overall survival was 100%, 83%, and 83% at 1-, 3-, and 5-years, respectively [46]. The patients had to show disease improvement or stability for 6 months before proceeding with transplantation, and choice of chemotherapy included gemcitabine/cisplatin or gemcitabine/capecitabine based on institutional protocols. Recurrent disease occurred in 50% of patients with a median time to recurrence of 7.6 months after transplantation.

Transplantation for iCCA remains a challenge as patient selection criteria are not well defined. Patients with small tumors, who respond to chemotherapy or have disease stability, and have a favorable tumor biology appear to benefit from liver transplantation. Further prospective protocols are needed to better define the ideal candidates for liver transplantation and choice of neoadjuvant therapy. There is currently a multicenter single-arm prospective trial (NCT02878473, https://clinicaltrials.gov/ct2/show/NCT02878473?cond=NCT02878473&draw=2&rank=1, accessed on 22 May 2021), started April 2018 and ongoing to confirm the utility of transplantation for single iCCA cases ≤ 2 cm in size.

### 3.3. Locoregional Therapies

Locoregional therapies (LRT), which include transcatheter arterial chemoembolization (TACE), yttrium-90 radio embolization (Y90-RE), and microwave and radiofrequency ablation, have been used in patients with iCCA with great success [47]. Chemotherapeutic agents like doxorubicin, cisplatin, mitomycin-C, and gemcitabine are administered during the TACE procedures with a carrier-like lipiodol followed by gelfoam, polyvinyl alcohol particles, or trisacryl gelatin (TG) microspheres to prevent the delivered agent from embolizing into the systemic circulation [47]. Survival rates with TACE are comparable to surgery in patients who did not achieve R0 resection [48].

During transarterial radioembolization (TARE) or Y90 procedures, the arterial supply to the carcinoma is identified via angiography, and small embolic particles (20–40 µm) that are either glass- or resin-based and are impregnated with the radio nucleotide Y90 are injected into the artery supplying the carcinoma. Y90 particles emit β-radiation, leading to local death of the malignant cells while sparing the rest of the healthy liver tissue; the Y90 procedure carries the advantage of a higher doses of radiation delivered directly to the tumor without systemic toxicity [49,50]. A systematic review and pooled analysis of 12 studies (*n* = 5 retrospective, *n* = 7 prospective) included 298 patients with iCCA who received Y90, had successful down staging and comparable overall survival to TACE and systemic chemotherapy [51].

Ablation via either chemical or thermal routes, the latter including radiofrequency ablation (RFA) and microwave ablation (MWA), has limited data in therapeutic management of iCCA associated with it compared to HCC, but a recent retrospective study showed that for non-surgical candidates, thermal ablation is a safe and effective treatment with good OS, however, it is associated with high rates of tumor recurrence. In the past, ablation was traditionally used as a palliative option in patients who were non-resectable and had small tumors (up to 5 total tumors, each less than 3 cm). Patients with single ≤ 2 cm lesions achieved good long-term survival when undergoing thermal ablation [52,53]. Zhang and colleagues showed that 1-, 3-, and 5-year OS rates were 93.5%, 39.6%, and 7.9%, respectively, with a median OS of 28.0 months in 107 patients with iCCA < 5 cm in size [54].

### 3.4. Chemotherapy

As discussed above, the majority of patients at initial presentation of iCCA are poor candidates for surgical resection, and in those who undergo surgical resection, recurrence rates are high with poor OS [16,55]. Therefore, the role of chemotherapy has been of great interest to improve OS and decrease disease recurrence. However, studies have shown mixed results for treatment of iCCA and are reviewed below with important studies summarized in Table 2.

### 3.5. Adjuvant Chemotherapy

Due to the low overall incidence of iCCA, evaluation of adjuvant chemotherapy in iCCA is often combined with remaining biliary tract cancers (BTC) despite significant differences in the biology of these tumors. BILCAP, a randomized phase 3 multicenter UK-based study, compared capecitabine therapy to observation after R0 (62%) or R1 (38%) resection of BTC and included 447 patients with 84 (19%) of those patients having iCCA [57]. While the intention-to-treat analysis was not statistically significant, per-protocol analysis revealed median OS at 53 months in the capecitabine arm compared to 36 months in the observation arm (HR 0.75, 95% CI 0.58–0.97; *p* = 0.028). In the intention-to-treat analysis, after adjusting for nodal status, tumor grade, and gender, there was an improvement in the OS in the capecitabine arm (HR 0.71, 95% CI 0.55–0.27; *p* = 0.010). Capecitabine is currently recommended as the first-line adjuvant chemotherapy after resection.

The PRODIGE 12-ACCORD 18 study, was a randomized phase 3 trial, where gemcitabine with oxaliplatin (GEMOX) was compared to surveillance in 196 patients with R0 or R1 resections for BTC, of whom 86 (44%) patients had iCCA, and failed to show a significant difference in relapse-free survival (RFS) or OS [58]. Retrospective studies have shown mixed results when evaluating the role of adjuvant therapy, but three recent systematic reviews and meta-analyses showed improvement in OS with adjuvant chemotherapy, supporting the additional role for chemotherapy compared to surgery alone [28,64,67]. A separate meta-analysis showed a non-significant benefit in OS for any adjuvant therapy compared to surgery alone, but when the role of adjuvant chemotherapy and radiation were compared, chemotherapy was statistically more beneficial than radiation therapy with the greatest benefit seen in patients with R1 resections and lymph node-positive disease [70]. Further studies are required to explore the benefits of various adjuvant chemotherapy options based on patient characteristics, tumor biology, and nodal status after resection of iCCA.

### 3.6. Neoadjuvant Chemotherapy

Data on neoadjuvant chemotherapy in iCCA remain limited to case series and retrospective studies with mixed results due to limited sample size and lack of control groups [71]. A recent propensity-matched retrospective review of 978 patients (71% with iCCA) from the National Cancer Database reported improved OS in patients receiving neoadjuvant chemotherapy before surgical resection [72]. There is growing interest in neoadjuvant therapy as evidenced by its role in multiple other cancers with the goal of down staging locally advanced disease and improving margin-negative resection. In a single-center retrospective French study [73], 186 patients with iCCA were reviewed with 90 patients deemed unresectable, 74 with locally advanced iCCA, and 16 with metastatic iCCA. Of the 74 patients with locally advanced cancers, all received neoadjuvant chemotherapy and 39 (53%) were able to undergo surgical resection with curative intent. The median OS for patients undergoing chemotherapy followed by resection was 3 years compared to 11 months in patients treated with chemotherapy alone (HR 4.58, 95% CI 2.59–8.09, *p* < 0.001). The neoadjuvant chemotherapy regimen was variable between both groups. Though this study supports the role of neoadjuvant chemotherapy as first-line treatment in locally advanced iCCA to down stage patients for surgical resection, further studies are needed to decide on the optimal chemotherapy regimen.

## 4. Advanced Intrahepatic Cholangiocarcinoma

Patients with unresectable or metastatic iCCA are considered to have advanced disease and carry poor prognosis with short OS with supportive care, while chemotherapy in these patients has shown improvement in their survival. The ABC-02, a phase 3 randomized clinical trial, compared cisplatin plus gemcitabine (GC) versus gemcitabine alone in 410 patients with locally advanced or metastatic BTC, of whom 80 (20%) patients had iCCA; the GC arm had better median OS (11.7 months vs. 8.1 months, HR 0.64, 95% CI, 0.52–0.80; *p* < 0.001) and improved progression-free survival (PFS) (8 months vs. 5 months, HR 0.63, 95% CI, 0.51–0.77; *p* < 0.001) compared to gemcitabine alone [59]. A Japanese phase 2 trial of 84 patients with advanced BTC, 28 (33%) of whom had iCCA, showed similar efficacy in median OS (11.2 months vs. 7.7 months) on GC therapy compared to gemcitabine alone, further supporting GC’s role in treating advanced iCCA [60]. Also, a post hoc analysis of ABC-01, -02, and -03 studies reviewing 109 patients with advanced iCCA, the majority (60.6%) of whom received GC, had better OS compared to other non-iCCA BTC [74]. Therefore, GC is the current first-line therapy recommended for patients with advanced non-resectable iCCA or metastatic BTC including iCCA.

In the ABC-06 randomized phase 3 trial of 162 patients with advanced BTC, of whom 72 (44%) had iCCA, patients who progressed on GC were treated with oxaliplatin plus fluoruracil (mFOLFOX) compared to supportive care alone and showed improved OS in the mFOLFOX arm [62]. Multiple phase 2 studies have shown the benefit of gemcitabine-based or fluoropyrimidine-based regimens as second-line agents in advanced BTC [75].

## 5. Targeted Therapy

Advances in DNA sequencing techniques, such as next generation sequencing (NGS), have allowed for simultaneous analysis of several genes or gene regions rapidly in a single test. Development of NGS in the last decade has allowed for molecular profiling of inherited or acquired mutations within multiple cancer types. Identifying these genomic mutations has led to the emergence of targeted therapies against molecular subtypes of cholangiocarcinoma with actionable genetic rearrangement in patients who have failed first-line therapies, with key mutations summarized in Table 3. Key oncogenic alterations include fibroblast growth factor receptor (FGFR) signaling and isocitrate dehydrogenase (IDH) mutations [76]. The FGFR family consists of four subtypes of transmembrane tyrosine kinase receptors, FGFR1–4, and the signaling pathway plays an important role in cellular proliferation, angiogenesis, survival, and migration; aberrant FGFR signaling promotes malignant transformation [77]. IDH is a citric acid cycle enzyme essential to cellular respiration and aerobic oxidation of pyruvate dehydrogenase; IDH has three subtypes, IDH1, IDH2, and IDH3. Mutations to IDH1 and IDH2 increase production of oncometabolite 2-hydroxyglutarate, leading to abnormal cellular proliferation and promotion of tumorigenesis [78].

Pemigatinib, a potent oral inhibitor of FGFR1, FGDFR2, and FGFR3, was evaluated in the FIGHT-202 trial, a multicenter open-label, single-arm phase 2 study that included 107 patients with advanced cholangiocarcinoma carrying FGFR2 abnormalities, of whom 89% had iCCA, showing an overall response rate of 36% and a median duration of response of 9.1 months [67]. A multicenter, randomized, double-blinded, placebo-controlled phase 3 study with a crossover design of 185 patients with advanced CCA (91% iCCA) was completed to assess the efficacy of ivosidenib (AG-120), a potent oral inhibitor of mutated IDH1 [67]. In the study’s primary end point, PFS was significantly improved with ivosidenib compared to the placebo, median duration of 2.7 months vs. 1.4 months (HR 0.37; 95% CI 0.25–0.54; *p* < 0.0001), with a non-significant favorable trend toward ivosidenib when assessing OS. Mismatch repair (MMR) deficiency has been identified in cholangiocarcinomas in small series [87], raising interest in the role of immune check point inhibitors, pembrolizumab, monoclonal antibody programed death receptor-1 (PD-1) inhibitors, with limited data currently available supporting their role in cholangiocarcinoma [64]. Nivolumab, a monoclonal antibody that blocks PD-1 and programed cell death 1 ligands, has been assessed in a multicenter phase 2 study of 54 patients with advanced BTC, 32 (59%) with iCCA, and revealed an objective response rate in 11% of patients [69]. Epidermal growth factor receptors (EGFR) are over-expressed in iCCA, and EGFR inhibitors such as erlotinib and cetuximab have been studied in combination with GEMOX with non-significant results [88]. Vascular endothelial growth factor (VEGF) inhibitors have been evaluated in advanced BTC, with bevacizumab studied in combination with GEMOX in a phase 2 study showing early promise [89], but the combination of an oral VEGF inhibitor, cediranib, with CisGem in the ABC-03 phase 2 trial failed to show improvement in PFS [63]. In addition, there has been growing interest in evaluating the role of germline mutations of BRCA1 and BRCA2 as recent studies have shown that these mutations confer an increased lifetime risk of developing CCA [90,91].

## 6. New Drugs on the Horizon

In an era of personalized medicine, tumor genomic profiling has opened the window to a better understanding of subtypes of iCCA while offering targeted therapy options. Targeted therapy involving IDH1 and IDH2 inhibitors, FGFR2 inhibitors, and human growth factor receptor (HER) inhibitors against advanced BTC are actively enrolling patients, while pembrolizumab is being studied in patients with MMR deficiency in all solid organ tumors, including BTC. While early EGFR and VEGF inhibitor studies have revealed mixed results, as discussed above, we anticipate that future therapies will adopt an individualized approach with molecular profiling and targeted therapies with a focus on combination therapies based on a patient’s CCA subtype.

## 7. Radiation

### 7.1. Adjuvant Radiation

A large retrospective analysis comparing surgery alone, surgery with adjuvant radiation, and radiation therapy alone as therapeutic options for iCCA revealed that adjuvant radiation followed by surgery resulted in the greatest OS benefit (HR = 0.40; 95% CI, 0.34–0.47), followed by surgery alone (HR, 0.49; 95% CI, 0.44–0.54), then radiation therapy alone (HR, 0.68; 95% CI, 0.59–0.77), when compared to no treatment [92].

### 7.2. Definitive Radiation

The combination of radiation with chemotherapy offers significantly higher survival than chemotherapy alone in unresectable iCCA [93]. In a study of 79 patients with inoperable iCCA with a median tumor size of 7.9 cm (range 2.2–17 cm) and where 89% received neoadjuvant chemotherapy followed by radiation, patients who received higher doses of radiation therapy had a better 3-year OS rate of 73% compared to patients receiving lower doses of radiation who had a 3-year OS rate of 38%; an improved 3-year local control rate of 78% was noted in patients receiving greater than 80.5 Gy of radiation when compared to those receiving lower doses (45%, *p* = 0.04) [94].

### 7.3. Proton Beam Therapy

There has been growing interest in utilizing proton beam therapy for the treatment of inoperable iCAA due to previous studies reporting excellent local control [95,96]. Study participants (*n* = 37) with biopsy-proven iCCA without extrahepatic disease and who were determined to be unresectable received 15 fractions of proton beam therapy to a maximum dose of 67.5 Gy. The median follow-up time was 19.5 months, the local control rate at 2 years was 94.1% with an OS of 46.5% [95].

### 7.4. Brachytherapy (BT)

In patients with unresectable BTC, a propensity-score matched-pair analysis showed that the addition of BT to radiation therapy (RT) showed better local control of the disease (35% (BT alone) vs. 65% (BT + RT), *p* = 0.094) but did not impact OS (31% (BT alone) vs. 40% (BT + RT), *p* = 0.862) [97].

## 8. Conclusions

Intrahepatic CCA is a rare malignancy that is difficult to treat with poor outcomes where the definitive treatment options, such as surgical resection and LT, are available to a small proportion of patients at initial presentation. Chemotherapy in combination with radiation have shown improvement in survival in patients with cholangiocarcinoma and a combination of Cis/Gem is currently the first-line treatment and standard of care for these tumors. Advances in genomic analysis and testing have expanded the opportunities for research and development of biomarker and targeted-therapy-driven trials. The technology of genomic profiling has improved our understanding of targeted therapeutic options, IDH1 and IDH2 inhibitors, FGFR2 inhibitors, and human growth factor receptor (HER) inhibitors. With all the new drugs being in various phases of clinical trial, we do anticipate that multiple options will be available for patients and combination therapies will offer better quality of life and prolonged survival. Locoregional options, including TACE, Y90, and ablation, help with local control of the disease. A multidisciplinary approach with a collaboration of various specialists, including hepatobiliary surgeons as well as medical, surgical, and radiation oncologists, has changed the landscape of the management and outcomes of this unique malignancy. This multidisciplinary approach is necessary given the high risk of recurrence and poor prognosis in operable cases, thereby raising the need for close monitoring and surveillance protocols. There remains a need to explore various combination regimens and prospective trials that focus on standardized selection criteria that can evaluate and develop therapies for iCCA patients.

## Figures and Tables

**Figure 1 jcm-10-02368-f001:**
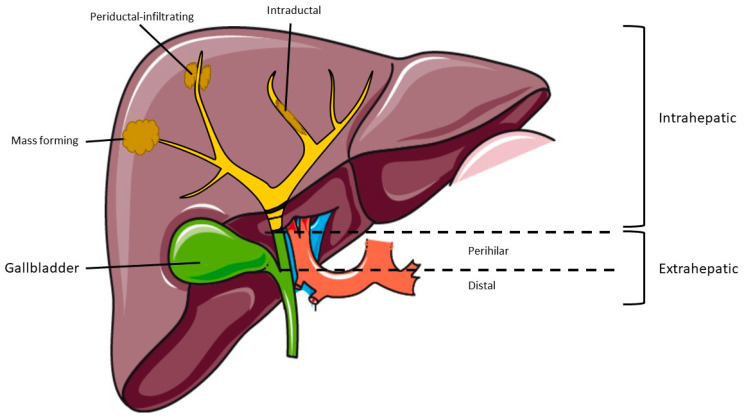
Classification of cholangiocarcinoma by anatomic location.

**Figure 2 jcm-10-02368-f002:**
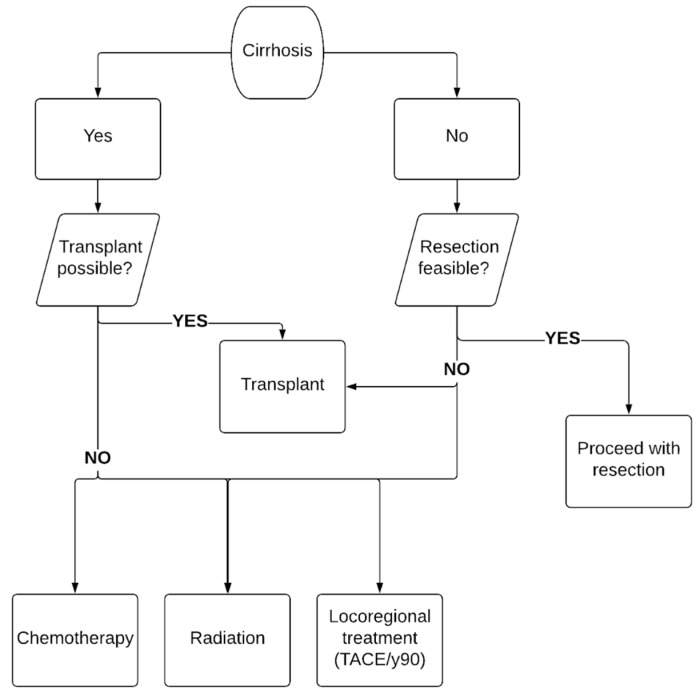
Approach to the management of cholangiocarcinoma.

**Table 1 jcm-10-02368-t001:** Major risk factors associated with the development of intrahepatic cholangiocarcinoma.

Risk Factors for iCCA
- Primary sclerosing cholangitis - Hepatolithiasis - Cirrhosis - Hepatitis B & C - Caroli’s disease - Liver flukes—*Clonorchis sinensis, Opisthorchis viverrini* - Obesity - Metabolic syndrome/non-alcoholic steatohepatitis - Alcohol consumption - Thorotrast

**Table 2 jcm-10-02368-t002:** Chemotherapy Regimens and Outcomes.

Author, Year, Study	Patients, Country	Location of BTC	Treatment Arm	()
**ADJUVANT THERAPY**				
Siebenhüner, 2018 Phase 2 [56]	30 Switzerland	Intrahepatic (57%) Extrahepatic (20%) Other (24%)	GC vs. gem	Median OS: no difference
Primrose, 2019 Phase 3 [57]	447 UK	Intrahepatic (19%) Extrahepatic (64%) Other (17%)	Capecitabine vs. surveillance	Median OS: 51.1 vs. 36.4 HR for OS: 0.71 (0.55–0.92) (*p* = 0.010) Median RFS: 24.4 vs. 17.5
Edeline, 2019 Phase 3 [58]	196 France	Intrahepatic (44%) Extrahepatic (36%) Other (20%)	GEMOX vs. surveillance	Median OS: no difference Median RFS: no difference
**THERAPEUTIC**				
Valle, 2010 Phase 3 [59]	410 UK	CCA * (59%) Other (41%)	GC vs. gem	Median OS: 11.7 vs. 8.1 HR for OS: 0.64 (0.52–0.80) (*p* < 0.001) Median PFS 8.0 vs. 5.0 (*p* < 0.001)
Okusaka, 2010 Phase 2 [60]	83 Japan	Intrahepatic (34%) Extrahepatic (23%) Other (43%)	GC vs. gem	Median OS: 11.2 vs. 7.7 Median PFS: 6.8 vs. 3.7 HR for PFS: 0.66 (0.41–1.05) (*p* = 0.077)
Kim, 2019 Phase 3 [61]	222 South Korea	CCA * (73%) Other (27%)	XELOX vs. GEMOX	Median OS: no difference ORR: no difference
Lamarca, 2019 Phase 3 [62]	162 UK	Intrahepatic (44%) Extrahepatic (28%) Other (28%)	Oxaliplatin + 5-FU + symptom control vs. symptom control	Median OS: 6.2 vs. 5.3 HR for OS: 0.69 (0.50–0.97) (*p* = 0.031)
**TARGETED THERAPY**				
Valle, 2015 Phase 2 [63]	124 UK	Intrahepatic (23%) Extrahepatic (39%) Other (38%)	Cediranib (anti-VEGF) + GC vs. GC	Median PFS: no difference
Abou-Alfa, 2020 Phase 2 [64]	146 Global	Intrahepatic (89%) Extrahepatic (8%) Other (3%)	Pemigatinib (anti-FGFR) + with or without FGF fusion/rearrangement	Median OS: 21.1 vs. 4.0 OR: 35.5% vs. 0% Median PFS: 6.9 vs. 1.7
Rizzo, 2020 Retrospective study [65]	450 Global	Intrahepatic (64%) Extrahepatic (18%) Other (18%)	EGFR inhibitor + gem vs. gem	Median OS: no difference Median PFS: no difference ORR: no difference
Ueno, 2020 Phase 2 [66]	26 Japan	Intrahepatic (23%) Extrahepatic (31%) Other (46%)	Lenvatinib (anti-VEGF)	Median OS: 7.36 ORR: 11.5% (3.2–27.2) Median PFS: 3.19
Abou-Alfa, 2020 Phase 3 [67]	185 Global	Intrahepatic (92%) Extrahepatic (3%) Other (5%)	Ivosideninb (IDH1 inhibitor) vs. placebo	Median PFS: 2.7 vs. 1.4 (*p* < 0.001) HR for PFS: 0.37 (0.25–0.54) (*p* < 0.0001) Median OS: no difference
**IMMUNOTHERAPY**				
Piha-Paul, 2020 Phase 1b, 2 [68]	104 (phase 2) 24 (phase 1b) Global		Pembrolizumab (anti-PD1)	KEYNOTE-158 (phase 2) Median OS: 7.4 Median PFS: 2.0 KEYNOTE-028 (phase 1b) Median OS: 5.7 Median PFS: 1.8
Kim, 2020 Phase 2 [69]	54 USA	Intrahepatic (59%) Extrahepatic (9%) Other (32%)	Nivolumab (anti-PD1)	Median OS: 14.24 Median PFS: 3.68 ORR: 22%

Abbreviations: Biliary tract cancer (BTC), cholangiocarcinoma (CCA), gemcitabine (Gem), gemcitabine + cisplatin (GC), gemcitabine + oxaliplatin (GEMOX), capecitabine + oxaliplatin (XELOX), gemcitabine + capecitabine (GemCap), gemcitabine+S1 (GS), overall survival (OS), progression-free survival (PFS), hazards ratio (HR), recurrence-free survival (RFS), objective response (OR), objective response rate (ORR), epidermal growth factor receptor (EGFR), fibroblast growth factor receptor (FGFR), vascular endothelial growth factor (VEGF). *, study did not differentiate between intrahepatic and extrahepatic cholangiocarcinoma.

**Table 3 jcm-10-02368-t003:** Overview of actionable genomic mutations identified in cholangiocarcinoma and potential therapeutic options.

Molecular Target	Frequency of Mutations (%)	Targeted Therapy
Cell cycle regulation		
TP53	12–43 [79,80]	p53 activators
CDKN2A	10–47 [81,82]	CDK 4/6 inhibitors
Epigenetic modification		
IDH1/2	3–36 [83,84]	IDH inhibitors
BAP1	4–32 [82,83]	PARP/ATM inhibitors
Chromatin remodeling		
ARID1A	5–27 [81,82]	EZH2, HDAC, DNMT inhibitors
PBRM1	6–21 [79,83]	
Kinase signaling		
KRAS	3–47 [79,85]	KRAS inhibitors
BRAF	1–45 [81,86]	BRAF inhibitors
EGFR	10–32 [84]	EGFR inhibitors
FGFR1/2/3	5–50 [17,84]	FGFR inhibitors
PIK3CA	5–13 [82,83]	mTOR inhibitors
c-MET	20–60 [84]	MET inhibitors

Abbreviations: CDKN2A, cyclin dependent kinase inhibitor 2A; IDH, isocitrate dehydrogenase; BAP1, BRCA1-associated protein 1; PARP, poly (ADP-ribose) polymerase; ATM, ataxia-telangiectasia mutated; ARID1A, AT-rich interaction domain 1A; PBRM1, protein polybromo-1; EZH2, enhancer of zeste homolog 2; DNMT, DNA methyltransferase; HDAC, histone deacetylase; EGFR, epidermal growth factor receptor; FGFR, fibroblast growth factor receptor; mTOR, mechanistic target of rapamycin; c-MET, mesenchymal-epithelial transition factor.

## Data Availability

Not applicable.

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
