# Peer review of "Management of Intrahepatic Cholangiocarcinoma"

_jcm, 2021, doi:10.3390/jcm10112368_

Round 1
Reviewer 1 Report
I would like to congratulate the authors on this comprehensive summary of the management options for intrahepatic cholangiocarcinoma.
Strength
Comprehensive summary of the literature and management options. It is useful to have a single document for up-to-date information about management across disease stages.
The comprehensive discussion about transplant
Opportunities to improve
Major
I think that the most interesting part of this review is the role of liver transplantation in the management of intrahepatic cholangiocarcinoma. You should consider moving this discussion up (and making it central)rather than leaving it towards the end. I do worry about the validity of Figure 2 though, I do not think that assessment for transplant eligibility is widely practised in the multidisciplinary discussions around ICC. If you are proposing that this should become more routine, then again the review would benefit from moving the discussion on transplant up.
Minor
While this was a comprehensive summary of the management of ICC, there wasn't a sufficient amount of critiquing the data and expert commentary on the direction the treatment options should be taking.
Author Response
Major
I think that the most interesting part of this review is the role of liver transplantation in the management of intrahepatic cholangiocarcinoma. You should consider moving this discussion up (and making it central)rather than leaving it towards the end. I do worry about the validity of Figure 2 though, I do not think that assessment for transplant eligibility is widely practised in the multidisciplinary discussions around ICC. If you are proposing that this should become more routine, then again the review would benefit from moving the discussion on transplant up.
Response: We appreciate your feed back. Have eloborated the liver transplant section and moved the section up to follow surgical resection. Also, the Figure 2 is what we propose and actually follow at our center and have made necessary edits to make that clear.
Minor
While this was a comprehensive summary of the management of ICC, there wasn't a sufficient amount of critiquing the data and expert commentary on the direction the treatment options should be taking.
Response: We have addended the conclusions section with some additional focus on molecular profiling and targetted therapies. Thank You
Reviewer 2 Report
This article represents an interesting overview of the iCCA treatment opportunities, which has been, in the past few years, a highly investigated topic. The paragraphs about surgical resection, locoregional therapies, adjuvant and neoadjuvant chemotherapy, radiation ad liver transplantation offer a clear representation of current opportunities. This is thanks to reporting and comparing different studies (some of which are still ongoing), which could be the starting point for future investigations. In addition, it also underlines the rising relevance of target therapy and the urge for more studies on this subject.
The weakness of the study is due to not taking into account data from a major study that shows how an increasing incidence of iCCA (see “Forty-Year Trends in Cholangiocarcinoma Incidence in the U.S.: Intrahepatic Disease on the Rise” by Supriya K Saha, Andrew X Zhu, Charles S Fuchs, Gabriel A Brooks).
The assessment of cancer incidence should also be based on different geographic areas as well as corresponding risk factors, all important elements for target therapy. Considering the risk factors, it might be interesting to explain the role of chronic inflammation, as the underlying process that mediates tumorigenesis, which is a major target for new drugs under investigation. Furthermore, it would be useful to consider an important study that showed different genetic mutations associated with different forms of biliary tract cancer, some of them typical of iCCA (see “Genomic spectra of biliary tract cancer” Hiromi Nakamura, Yasuhito Arai, Yasushi Totoki, et al.
Also, this review lacks some other relevant review on the same issue, such as “New and Emerging Systemic Therapeutic Options for Advanced Cholangiocarcinoma” by S Massironi, L Pilla, A Elvevi, et al. Cells. 2020 Mar 11;9(3):688. doi:10.3390/cells9030688.
Finally, the article has an appropriate length, nevertheless, the introduction and the paragraph about diagnosis are written in less formal English than the rest of the article.
Author Response
This article represents an interesting overview of the iCCA treatment opportunities, which has been, in the past few years, a highly investigated topic. The paragraphs about surgical resection, locoregional therapies, adjuvant and neoadjuvant chemotherapy, radiation ad liver transplantation offer a clear representation of current opportunities. This is thanks to reporting and comparing different studies (some of which are still ongoing), which could be the starting point for future investigations. In addition, it also underlines the rising relevance of target therapy and the urge for more studies on this subject.
Response: Thank You very much for your positive comments.
The weakness of the study is due to not taking into account data from a major study that shows how an increasing incidence of iCCA (see “Forty-Year Trends in Cholangiocarcinoma Incidence in the U.S.: Intrahepatic Disease on the Rise” by Supriya K Saha, Andrew X Zhu, Charles S Fuchs, Gabriel A Brooks).
Response: The study by Saha et al was used in the submitted manuscript but we have cited their study in further detail now describing the trends and changing incidence of iCCA.
The assessment of cancer incidence should also be based on different geographic areas as well as corresponding risk factors, all important elements for target therapy. Considering the risk factors, it might be interesting to explain the role of chronic inflammation, as the underlying process that mediates tumorigenesis, which is a major target for new drugs under investigation. Furthermore, it would be useful to consider an important study that showed different genetic mutations associated with different forms of biliary tract cancer, some of them typical of iCCA (see “Genomic spectra of biliary tract cancer” Hiromi Nakamura, Yasuhito Arai, Yasushi Totoki, et al.
Response: We have expanded on the role of chronic inflammation in tumorigenesis and have further described the genetic mutations by citing the above mentioned study by Nakamura etal.
Also, this review lacks some other relevant review on the same issue, such as “New and Emerging Systemic Therapeutic Options for Advanced Cholangiocarcinoma” by S Massironi, L Pilla, A Elvevi, et al. Cells. 2020 Mar 11;9(3):688. doi:10.3390/cells9030688.
Response: We have also expanded on new and emerging options and used the above review by Massironi et al.
Finally, the article has an appropriate length, nevertheless, the introduction and the paragraph about diagnosis are written in less formal English than the rest of the article.
Response: we have attempted to rephrase/ rewrite portions of intorduction and Diagnosis sections as well. Appreciate your time and valuable input.